# If You've Trained One You've Trained Them All: Inter-Architecture Similarity Increases With Robustness

**Haydn T. Jones**[1]     **Jacob M. Springer**[1]     **Garrett T. Kenyon**[1]     **Juston S. Moore**[1]

[1]Los Alamos National Laboratory, Los Alamos, New Mexico, United States

## Abstract

Previous work has shown that commonly-used metrics for comparing representations between neural networks overestimate similarity due to correlations between data points. We show that intra-example feature correlations also causes significant overestimation of network similarity and propose an image inversion technique to analyze only the features used by a network. With this technique, we find that similarity across architectures is significantly lower than commonly understood, but we surprisingly find that similarity between models with different architectures increases as the adversarial robustness of the models increase. Our findings indicate that robust networks tend toward a universal set of representations, regardless of architecture, and that the robust training criterion is a strong prior constraint on the functions that can be learned by diverse modern architectures. We also find that the representations learned by a robust network of any architecture have an asymmetric overlap with non-robust networks of many architectures, indicating that the representations used by robust neural networks are highly entangled with the representations used by non-robust networks.

## 1 INTRODUCTION

There is evidence that neural networks—across architectures and weight initializations—rely on similar features for classification. Previous literature has proposed the *universality hypothesis*, which posits that neural networks learn essentially the same representations when trained on the same data, regardless of exact architecture or training algorithm [Olah et al., 2020]. However, we know that different

Code to produce inversions can be found in our accompanying blog post: haydn-jones.github.io/posts/a-better-index-of-similarity

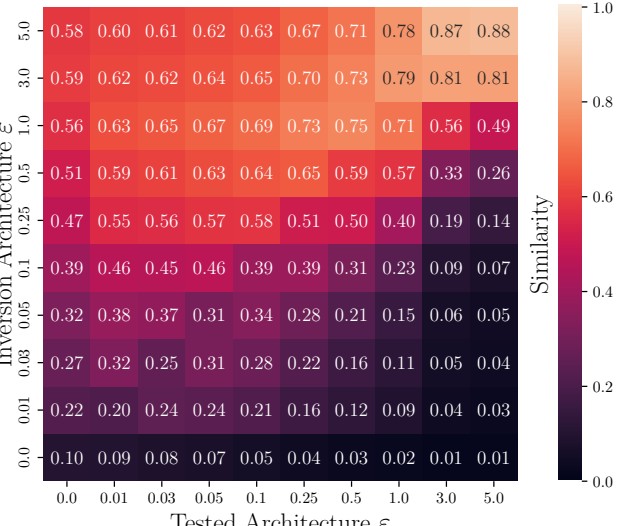

Figure 1: Representation-layer similarity of neural networks increases with robustness. Our proposed metric, based on image inversions, is averaged across every pair of architectures. The representations used by robust neural networks are extremely similar across architectures and random initializations, and show high similarity with non-robust networks.

architectures and random initializations often make different predictions [Lakshminarayanan et al., 2017]. This paper re-examines the similarity question from a novel viewpoint by considering the effect of an adversarial robustness constraint during training.

Robust training is a well-established procedure to decrease the sensitivity of a network's outputs to small changes in inputs Madry et al. [2018]. It is well-known that increasing the robustness of neural networks against adversarial examples comes with a cost to accuracy [Tsipras et al., 2019]. However, little attention has been paid to the effect that robust training has on agreement between models. We empirically show through multiple methods of similarity analysis that the representations, and consequently the functions, learned

*Accepted for the 38th Conference on Uncertainty in Artificial Intelligence* (UAI 2022).

by networks of different architectures become significantly more similar as robustness increases. This finding indicates that robustness serves as a strong prior on the functions that can be learned, and in fact may be a strong enough constraint to matter more than the specific network architecture. While surprising, this hypothesis is consistent with recent theoretical results showing that modern networks are actually under-parameterized to represent smooth functions in high dimensions [Bubeck and Sellke, 2021].

Many methods have been proposed to measure similarity between neural networks, including centered kernel alignment (CKA) [Kornblith et al., 2019], Canonical Correlation Analysis (CCA) [Hardoon et al., 2004], singular vector canonical correlation analysis (SVCCA) [Raghu et al., 2017], subspace match [Wang et al., 2018] and more. In this paper, we show that existing methods for evaluating representation similarity tend to over-estimate similarity due to feature correlations. We develop a novel method for measuring network representation similarity that deconfounds the effect of correlated features by constructing model-specific datasets where each data point has been transformed to contain only the features used by the model. Using this metric, we present a comprehensive study examining the similarity between neural networks as a function of the robustness level used in adversarial training.

Figure 1 summarizes our results using the novel similarity metric we propose. The similarity between networks with different architectures is extremely high amongst robust networks, and robust networks indeed show high similarity with non-robust networks, indicating that there is significant entanglement between robust and non-robust representations. Overall, we find two novel and surprising results:

1. Similarity between networks trained with empirical risk minimization is limited, and this similarity is overestimated by existing correlation-based similarity metrics due to feature correlations present in the dataset itself. We present experimental evidence that non-robust neural networks rely on features that are highly correlated in the data yet measure distinct patterns.

2. Adversarial robustness is a strong constraint on the function that is learned by a network, regardless of architecture or random initialization. We find that similarity between robust networks is extremely high, and that robust networks also show asymmetric similarity with non-robust neural networks.

## 2 RELATED WORK

Cui et al. [2022] investigate the confounding effect of input data on representational similarity across deep neural networks. They highlight that *inter*-example similarity can cause representation similarity to be spuriously high even between networks with Gaussian noise added to their pa-

rameters. They propose to regress out the input similarity structure from the representation similarity structure and find that doing so corrects for the failure of CKA to distinguish between random neural networks, among other benefits. Ding et al. [2021] further investigates issues surrounding similarity indices, finding that CKA and CCA both fail to satisfy at least one of their proposed criteria expected of similarity metrics. We identify an additional limitation to these similarity metrics by showing that correlated features in the input data can likewise cause the similarity between neural networks to be overestimated.

Bai et al. [2021] propose a method to study the representations learned by neural networks by removing all non-linear components of a network and integrating all linear components into linear subnetworks $W$. By analyzing the weight vectors of $W$, the authors find that adversarially-trained networks cluster along class hierarchies while standard networks do not. Likewise, Salman et al. [2020] find that the representations learned by robust neural networks provide a better starting point for transfer learning than the representations learned by non-robust networks. These findings may indicate that adversarially-trained networks are extracting more semantic and generalizable representations than standard networks.

Springer et al. [2021b] investigate the ability of adversarially trained neural networks to generate targeted adversarial examples. They find that classifiers that have been adversarially trained, even those only robust to small-magnitude perturbations, are much more effective than standard classifiers at generating targeted adversarial examples. Based on their findings, they argue that the representations used by slightly robust neural networks are shared widely across non-robust networks. We present further evidence for this hypothesis by evaluating the representational similarity of robust and non-robust neural networks.

## 3 METHODS

We consider a labeled classification dataset $\mathcal{D} = X, y$ of data points and ground truth labels. Given two neural networks $f_1$ and $f_2$, comprised of layers

$$f_1 = f_1^{(L_1)} \circ f_1^{(L_1-1)} \circ \ldots f_1^{(1)}$$
$$f_2 = f_2^{(L_2)} \circ f_2^{(L_2-1)} \circ \ldots f_2^{(1)}$$

we are interested in the activations at the *representation* layers (i.e. the second-to-last, or penultimate, layer) of $f_1$ and $f_2$, denoted $A$ and $B$

$$g_1 = f_1^{(L_1-1)} \circ \ldots \circ f_1^1 \qquad A = g_1(X)$$
$$g_2 = f_1^{(L_1-1)} \circ \ldots \circ f_1^1 \qquad B = g_2(X)$$

## 3.1 CENTERED KERNEL ALIGNMENT

We use centered kernel alignment to compare representations between neural networks. Given two mean-centered matrices of activations $A \in \mathbb{R}^{n \times p_1}$ and $B \in \mathbb{R}^{n \times p_2}$ of $p_1$ and $p_2$ neurons on a set of $n$ examples, CKA computes a value in the range $[0, 1]$ with values closer to 1 indicating higher similarity. Kornblith et al. [2019] show that CKA has a number of desirable properties, including the ability to calculate similarity between layers with a different numbers of neurons, invariance to isotropic scaling, and the ability to identify correspondences between the layers of identical architectures trained from different initializations—properties many widely used metrics lack.

Since Kornblith et al. [2019] show that linear and radial basis function kernels compute similar similarity indices, we use a linear kernel for simplicity. The linear CKA between $A$ and $B$ is given by:

$$
\begin{aligned}
\mathrm{CKA}(A, B) &= \frac{\|B^T A\|_F^2}{\|A^T A\|_F \|B^T B\|_F} \\
&= \frac{\|cov(A^T, B^T)\|_F^2}{\|cov(A^T, A^T)\|_F \|cov(B^T, B^T)\|_F}
\end{aligned}
\tag{1}
$$

where $\|\cdot\|_F$ denotes the Frobenius norm. Previous studies have evaluated CKA between each pair of layers in the networks to be compared. In this study, we restrict our attention to the penultimate layer of each network, since this layer effectively captures the summary statistics used by each network for classification.

In this work, we empirically demonstrate a shortcoming of CKA, as traditionally applied. Suppose that two features are perfectly correlated in both the train and test partitions of a dataset: $X_{.,i} = c\, X_{.,j}$ for some $|c| > 0$ and $i \neq j$. Also suppose that each network computes its representations $A_{.,l}$ and $B_{.,m}$ using only one of these correlated features: $i$ for $f_1$ and $j$ for $f_2$, respectively. Then, CKA evaluated on test data will show that representations $l$ and $m$ are perfectly correlated, although it does not indicate similar feature usage between the networks. In large, high-dimensional datasets with many correlated features, such as natural images, this shortcoming can lead to a dramatic overestimation of network similarity.

## 3.2 REPRESENTATION-LAYER INVERSION.

We use an inversion technique from the adversarial robustness literature to eliminate the effect of correlated features in test data. Ilyas et al. [2019] proposed *representation inversion* to investigate the features used by robust and non-robust classifiers. Representation inversion constructs a model-specific *inverted dataset* $\mathcal{I}_f(\mathcal{D})$ for a given classifier $f$ where all features *not* used by $f$ are randomized. This technique was developed in the computer vision domain, and our experiments use images, so we will explain repre-

sentation inversion in terms of images; note, however, that the inversion process is generally applicable to datasets with continuous-valued features.

Given a labeled dataset $\mathcal{D}$, we choose a pair of inputs that have *different* labels. The first of each pair will be the *seed* image $s$ and the second the *target* image $t$. Using the seed image as a starting point, we perform gradient descent to find an image that induces the same activations at the representation layer as the target image, under a specific neural network $f$. We construct this image through gradient descent in input space (with the constraint that the resulting image has pixel values in the range $[0, 1]$) by optimizing the following objective:

$$
\tilde{s} = \min_s \frac{\|g(s) - g(t)\|_2}{\|g(t)\|_2}
\tag{2}
$$

By sampling pairs of seed and target images that have distinct labels we eliminate features correlated with the target class that are not used by the model for classification. For example, if we have a seed and target images of "microwave" and "cat" respectively, and a given neural network is using the feature of "cat ear" to classify cat images but *not* "cat fur", by starting from an image of a microwave we can produce an image with the model's representation of "cat ear" that does not contain the correlated feature of "cat fur". If done successfully, the similarity between a neural network relying only on "cat fur" and a neural network relying only on "cat ear" will not be inflated by the co-occurrence of these features in the original dataset.

## 3.3 REPRESENTATION STITCHING

Csiszárik et al. [2021] approach neural network similarity from the *functional* perspective, asking the question "*can network $f_2$ achieve its task using only the representations of network $f_1$?*". Their simple and elegant method stitches together the activations $A$ from a *body* network $f_1$ with the last layer of a *head* network $f_2^{(L_2)}$ by fitting an affine transformation to match the activations of the head network: $B \approx AW + b$. This procedure creates a *stitched* network:

$$
f_{2 \circ 1} = f_2^{(L_2)} \circ (g_1 W + b)
$$

We construct stitched networks by fitting $W, b$ via least squares on activations of the body and head models computed on training data, and we perform no task-specific fine tuning. If there exists an identifiable linear transformation between the networks at the penultimate layer, then the stitched network will achieve high performance. Importantly, since the last layer of each network is a dense layer followed by a softmax, when the representation stitching procedure causes the stitched model to agree with the head model, it shows that the representations computed by the body network are compatible and useful with respect to the head model.

| $\varepsilon$ | 0 | .01 | .03 | .05 | .1 | .25 | .5 | 1 | 3 | 5 |
|---|---|---|---|---|---|---|---|---|---|---|
| ResNet18 | ✓ | ✓ | ✓ | ✓ | ✓ | ✓ | ✓ | ✓ | ✓ | ✓ |
| ResNet50 | ✓ | ✓ | ✓ | ✓ | ✓ | ✓ | ✓ | ✓ | ✓ | ✓ |
| WRN50-2 | ✓ | ✓ | ✓ | ✓ | ✓ | ✓ | ✓ | ✓ | ✓ | ✓ |
| WRN50-4 | ✓ | ✓ | ✓ | ✓ | ✓ | ✓ | ✓ | ✓ | ✓ | ✓ |
| ResNeXt50 | ✓ | ✗ | ✗ | ✗ | ✗ | ✗ | ✗ | ✗ | ✓ | ✗ |
| VGG16-bn | ✓ | ✗ | ✗ | ✗ | ✗ | ✗ | ✗ | ✗ | ✓ | ✗ |
| DenseNet | ✓ | ✗ | ✗ | ✗ | ✗ | ✗ | ✗ | ✗ | ✓ | ✗ |
| ShuffleNet | ✓ | ✗ | ✗ | ✗ | ✗ | ✗ | ✗ | ✗ | ✓ | ✗ |
| MobileNet | ✓ | ✗ | ✗ | ✗ | ✗ | ✗ | ✗ | ✗ | ✓ | ✗ |

Table 1: Pretrained ImageNet models used in experiments with available $\ell_2$ robustnesses, provided by Salman et al. [2020] (github.com/microsoft/robust-models-transfer). WRN50-$N$ is a WideResNet50-$N$ [Zagoruyko and Komodakis, 2016].

## 3.4  ADVERSARIAL TRAINING

Adversarial training has been shown to be effective for constructing neural networks that are robust to adversarial examples [Madry et al., 2018]. In addition, adversarial training yields neural networks with a number of desirable qualities, including interpretable gradients [Tsipras et al., 2019], high-quality representations that are useful for transfer learning [Salman et al., 2020], and the ability to generate transferable adversarial examples [Springer et al., 2021c]. Despite the extensive research into this training paradigm, to our knowledge no comprehensive study has explored the relationship between network similarity and robustness.

During adversarial training, the empirical risk minimization regime is changed to a min-max criterion to produce a model that is robust to adversarial perturbations within a bounded region $S(x)$ around each training point $x$. We use the common choice of an $\varepsilon \in \mathbb{R}$ sized $\ell_2$-ball and refer to $\varepsilon$ as the *robustness level* of the model. The training loss for robust training is:

$$\min_{\theta} \; \mathbb{E}_{(x,y)\sim\mathcal{D}} \left[ \max_{\delta \in S(x)} \mathcal{L}(x+\delta, y; \theta) \right] \qquad (3)$$

Due to computational requirements, we use the pretrained $\ell_2$-robust ImageNet models released by Salman et al. [2020]. The specific architectures and robustnesses ($\varepsilon$) studied are outlined in Table 1.

## 4  EXPERIMENTS

We evaluate our proposed method for estimating representation similarity, and show that it leads to consistent conclusions with other accepted methods for network similarity evaluation. Section 4.2 discusses the overestimation of neural network similarity and proposes a novel method for similarity estimation based on image inversions. Sections 4.3

to 4.5 discuss our findings on the convergence of representation similarity across robust neural networks, demonstrating that disparate methods for similarity estimation lead to the consistent conclusion that similarity increases significantly between architectures and random initialization as a function of robustness.

## 4.1  DATASET CONSTRUCTION

We perform the representation-layer inversion process on a subset of $10,000$ images drawn from the ImageNet validation set [Deng et al., 2009], producing a unique inverted dataset $\mathcal{I}_f(\mathcal{D})$ for each model. There is no constraint to limit the difference between the seed image $s$ and the inverse image $\tilde{s}$, and the inversion is performed to match *activations*, not just classification outputs. Thus our approach is fundamentally different from presenting a network with adversarial examples. We found in our experiments that gradient descent finds inverse images with representations very closely matching the target images, regardless of the seed/target pair chosen. The distance between the seed image $s$ and the inverted image $\tilde{s}$ tends to increase as the robustness of the network $f$ increases (Figure 2).

To produce the inverted datasets, $\mathcal{I}_f(\mathcal{D})$, we use PyTorch [Paszke et al., 2019] and the Robustness library [Engstrom et al., 2019], minimizing the objective defined in Equation (2). *All models are set to evaluation mode before starting the inversion process*. For faster convergence we implement $\ell_2$ momentum as described in Dong et al. [2018]. Inversions are produced through $2,000$ steps of gradient descent with a step size of $1/8$ and $\ell_2$ momentum of $0.9$. For each inversion we choose the point along the optimization path that minimizes the objective Equation (2) rather

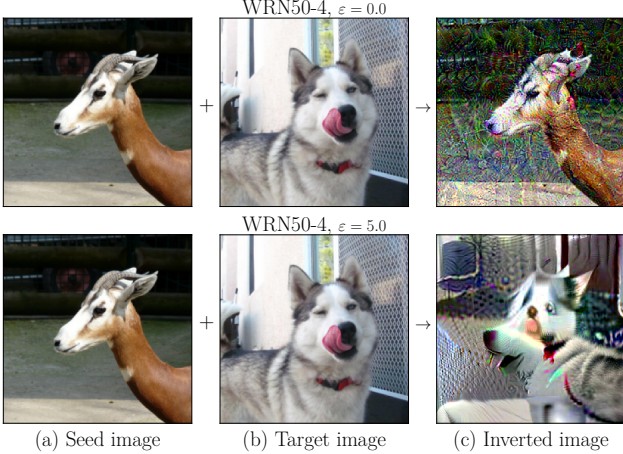

WRN50-4, $\varepsilon = 0.0$

WRN50-4, $\varepsilon = 5.0$

(a) Seed image  (b) Target image  (c) Inverted image

Figure 2: Inversion process example for a standard WRN50-4 (top row) and a WRN50-4 that was trained with $l_2$ robustness $\varepsilon = 5.0$ (bottom row). The target images and inverted images induce the same activations at the representation layer of the network despite being visually distinct.

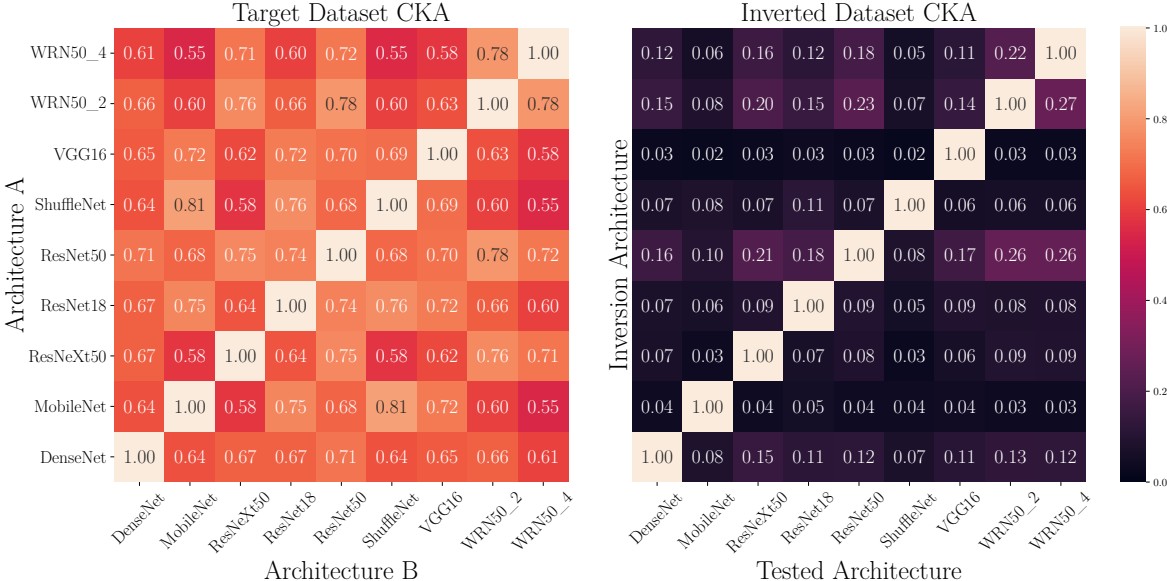

Figure 3: Representation layer similarity of *non-robust* neural networks on the test set $\mathcal{D}$ and the inverted datasets $\mathcal{I}_f(\mathcal{D})$. High similarity is observed under the natural image dataset and low similarity under datasets generated by inverting non-robust models. These results indicate that similarity is systematically overestimated when based on responses to natural images but can be more reliably estimated using responses to inverse images.

than the final result of gradient descent. At each step we normalize the gradient to unit norm.

## 4.2 OVERESTIMATION OF NEURAL NETWORK SIMILARITY

Many previous studies have examined the similarity between neural networks with different architectures or weight initializations [Kornblith et al., 2019, Nguyen et al., 2021, Hermann and Lampinen, 2020]. Here, we argue that similarity between networks may be substantially overestimated by these studies. While these studies suggest that neural network activations are often highly correlated across different networks, none of them have taken into account the substantial confounder that neuron responses may appear to be correlated despite responding to distinct patterns, due to frequent co-occurrence of the patterns in the dataset.

We first present experimental results demonstrating that this confounder is present for standard, *non-robust* neural networks, shown in Figure 3. The left heatmap presents the CKA similarity at the representation layer between all non-robust architectures on a subset of the ImageNet validation set. Note that CKA is symmetric, so the plot on this natural dataset is symmetric. As is commonly reported, the similarity between all architectures is relatively high with an average of 0.67 between the penultimate layers of distinct architectures. In the right heatmap, at each row-column entry, we present the CKA similarity between the row and column architecture using the inverted dataset $\mathcal{I}_f(\mathcal{D})$ generated by the *row*'s architecture. Because the dataset varies by row,

this plot is not symmetric. In contrast to the high similarities found in the left heatmap, all similarities are significantly lower with an average between distinct architectures of 0.09.

The trends shown in Figure 3 clearly demonstrate that current similarity metrics are overestimating network similarity to a significant degree due to correlations between distinct features in the data. When images containing only the relevant features for one of the models are used in the similarity calculation, we see that models are far more dissimilar than standard metrics indicate. We therefore propose evaluating CKA on inverted image datasets in order to best measure the manner in which each network computes sufficient and necessary features for classification.

## 4.3 ADVERSARIAL TRAINING INCREASES REPRESENTATION SIMILARITY

We apply our new method for network similarity estimation to investigate how robustness affects the similarity of neural networks. In Figure 4 we calculate similarity between networks in a similar fashion to Figure 3; however, this time all architectures being compared were adversarially trained with an $\ell_2$ robustness of $\varepsilon = 3$. In this plot we find that robust networks are significantly more similar to each other than non robust networks are on *both* the natural dataset and the inverted datasets $\mathcal{I}_f(\mathcal{D})$. On the natural dataset, robust models have an average similarity of 0.83 compared to 0.67 for standard models. On $\mathcal{I}_f(\mathcal{D})$, robust models have an average similarity of 0.80, compared to 0.09 for standard models as seen in Figure 3. This may indicate that robust

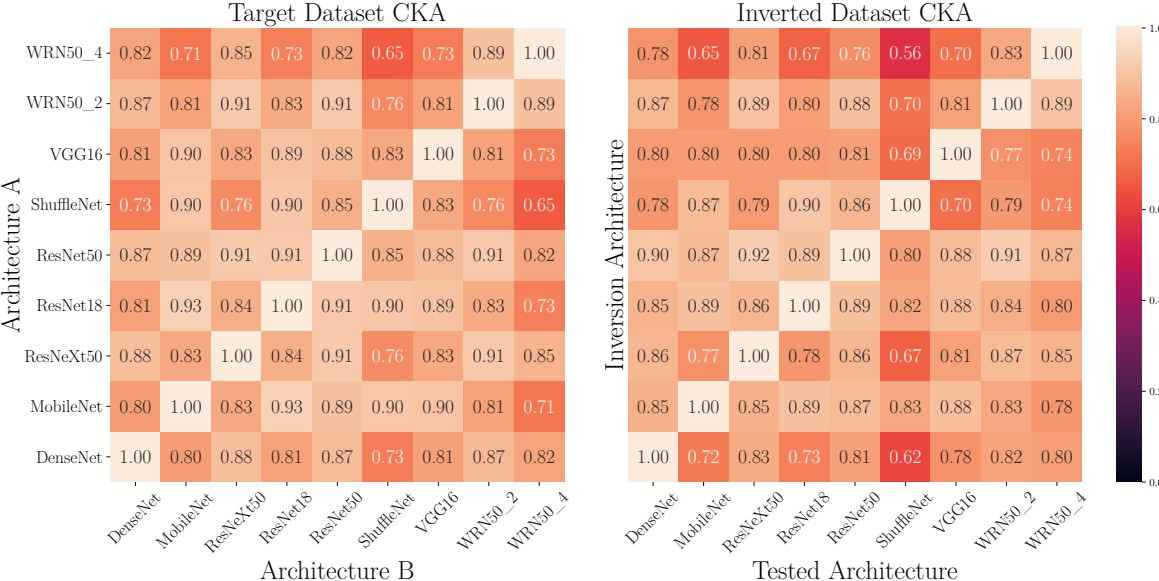

Figure 4: Representation layer similarity of robust neural networks ($\varepsilon = 3$) on the test set $\mathcal{D}$ and the inverted datasets $\mathcal{I}_f(\mathcal{D})$. Similarity is much higher than observed in Figure 3 on *both* the inverted datasets and the natural datasets. The fact that similarity measured via standard CKA (left) and our method (right) is very close indicates that highly robust networks of different architecture are indeed using very similar features to perform classification.

neural networks tend towards a similar set of representations, regardless of architecture.

In Figure 6 we plot the similarity between neural networks across architecture and robustness on the inverted datasets $\mathcal{I}_f(\mathcal{D})$, restricted to a comparison of three architectures for readability. In each heatmap we present the similarity between the outer-row and outer-column architecture, varied across $\varepsilon$ for each. Each inner-row and inner-column tick corresponds to the robustness of their respective outer-row and outer-column architecture, with the row architecture being the source of the inverted dataset $\mathcal{I}_f(\mathcal{D})$. In these plots we see a strong and consistent trend: when similarity is calculated between architectures using an inverted dataset produced by a non-robust or slightly-robust model (low value column ticks), low similarity is observed across all target architecture (i.e. column) robustnesses. However, when an inverted dataset produced by a *robust* architecture is used, we see notably *higher* similarity across all target architecture robustnesses. Average comparisons between all architectures are shown in Figure 1, showing that this trend holds for all architectures evaluated.

Our results show a strong asymmetric relationship between robust and non-robust models. Inversions produced by a robust model show high similarity with all other models, but inversions produced by a non-robust model show low similarity with all other models. This finding supports the idea that features used by non-robust neural networks are highly entangled with the features used by robust neural networks [Springer et al., 2021a]. As is seen in Figure 6, the features present in the inverted datasets $\mathcal{I}_f(\mathcal{D})$ generated by

robust models are causing the features used by non-robust classifiers to activate, yet the opposite does not hold.

## 4.4 ADVERSARIAL TRAINING INCREASES GRADIENT SIMILARITY

We further show the increase in similarity among robust neural networks with an intriguing result that as the pairwise-robustness of models increase, the cosine similarity of their saliency maps [Simonyan et al., 2014] with respect to the ground truth labels increases as well. For each pair of models at identical robustness, we compute the cosine similarity between their saliency maps, and present the results in Fig-

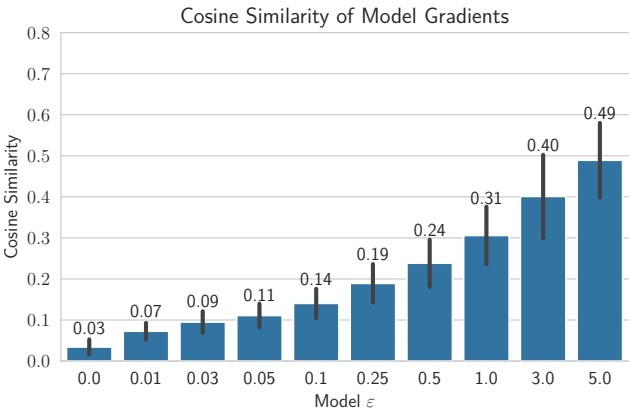

Figure 5: Cosine similarity of model gradients in input space across robustness levels. Error bars indicate 1 std. deviation.

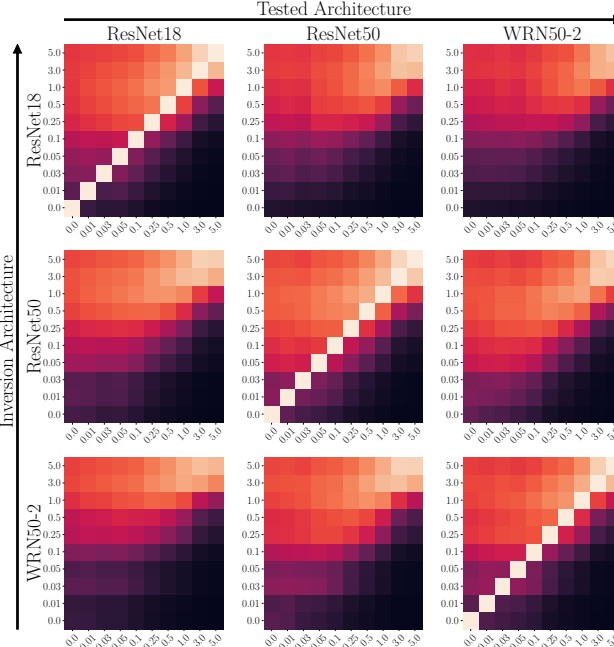

Figure 6: Representation-layer similarity of neural networks on the inverted datasets, $\mathcal{I}_f(\mathcal{D})$, pair-wise across different robustness levels.

ure 5. In a similar fashion to the results in Section 4.2 and Section 4.3, we find that the similarity between standard models is quite low, however, as the pairwise robustness between models increases, so does their gradient similarity.

## 4.5 ADVERSARIAL TRAINING INCREASES FUNCTIONAL SIMILARITY

It has been argued that assessments of network similarity must examine *functional similarity*, since evaluation on known ground-truth labels gives a concrete, performance-based test [Ding et al., 2021]. Here, we apply the representation stitching methodology (Section 3.3) to "stitch" together networks of two different architectures using a single affine transformation. Within each robustness level, we evaluate every pair of models, using the first model as a *body* and the second model as a *head*. When the head and body models disagree, evaluating the agreement of the stitched model with the head model can loosely be thought of as evaluating whether the linear classifiers of the head and body models are using the same set of features in different proportions to produce their classifications. If this is in fact the case, by mapping the body features into the linear subspace of the head model, one should expect an increase in agreement of the head model with the stitched model.

Figure 7 shows that stitching representations across architectures is effective; regardless of robustness level, when the head and body networks agree on a label, the stitched model also predicts the same label (93.9% for correct labels and 83.2% for incorrect labels). Interestingly, when the head and body networks predict different labels, we find that agreement between the stitched model and the head model increases significantly as a function of robustness (Figure 8). In other words, it appears that there is a stronger linear correspondence between representations of the body and head network, from a functional perspective, at higher robustness levels.

## 5 DISCUSSION

We find that while existing correlation-based similarity metrics overestimate the similarity between non-robust neural networks due to co-occurrence of features in the evaluation dataset, robust neural networks exhibit substantial similarity. We find that as the adversarial robustness of a neural network increases, its similarity to other networks increases, even across differences in architecture and random initialization. This trend of similarity is also present in the gradients, where we find that the Jacobians with respect to inputs are more similar for robust networks than for their non-robust counterparts. These results suggest a modified universality hypothesis, which suggests that neural networks, regardless of exact training condition (i.e., architecture, random initialization, learning parameters) will learn similar representations under mild constraints, such as adversarial robustness. We find empirically that robust neural networks satisfy this hypothesis. Furthermore, we find that the representations of non-robust neural networks overlap substantially with the representations of robust neural networks despite less overlap with the representations of other non-robust neural networks. This suggests that non-robust representations can be thought of as "components" of robust representations, much as a feature that represents the ear of a cat can be thought of as a component of the feature that represents an entire cat.

Our results provide an important step towards understanding the representations learned by neural networks. Our framework justifies the previously observed exceptional transferability of adversarial examples constructed using robust neural networks by demonstrating that non-robust and robust representations exhibit substantial overlap [Springer et al., 2021b]. In addition, we suspect that the fact that a single robust neural network has some degree of similarity to all non-robust neural networks can explain why robust neural networks are often better at learning representations that transfer to new tasks [Salman et al., 2020].

Neural network architectures and optimization procedures have often been viewed from a Bayesian perspective as a strong prior on the functions that these networks can learn [Wilson and Izmailov, 2020, Kleinberg et al., 2018]. Our results indicate that robust training, likewise, is a very strong prior which can constrain both the representations extracted

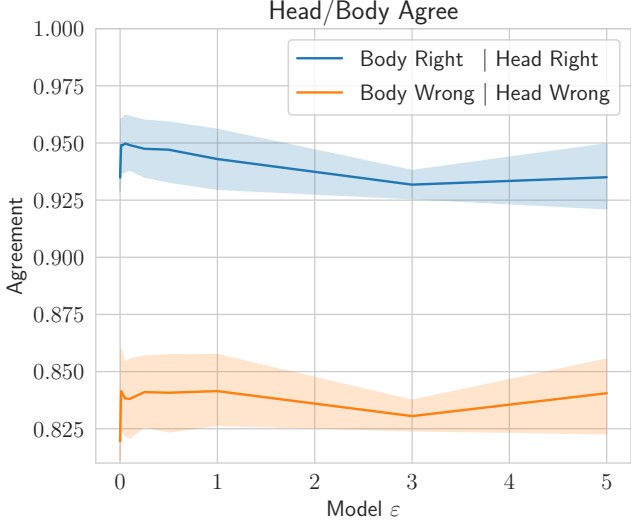

Figure 7: Average agreement between stitched models and their corresponding head models on data instances where the body and head networks agree. Robustness has no effect on agreement. Confidence bands indicate 1 std. error.

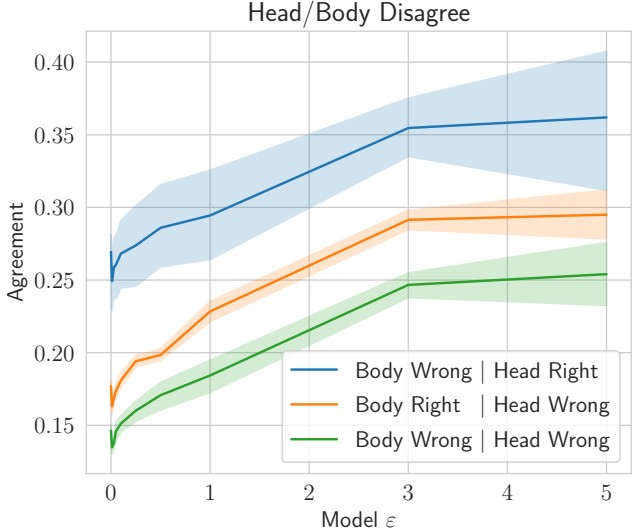

Figure 8: Average agreement between stitched models and their corresponding head models on data instances where the body and head networks disagree. Agreement increases as robustness increases. Confidence bands indicate 1 std. error.

from a dataset and the functions learned, independent of architecture. Viewing adversarial robustness as an inductive bias may lead to understanding the limitations of our current models for adversarial robustness, and may help us develop better notions of robustness and improve the accuracy of robust models.

# 6 CONCLUSION

Increased similarity between robust neural networks could mean that empirical analysis of a single robust neural network will reveal insight into the representations learned by *every* other robust neural network, which may lead us to understand the nature of adversarial robustness itself. If neural networks learn a solution that is largely dependent on the data itself rather than the learning algorithm, random initialization, or architecture, then we may be able to derive insight into the innate structure of data by using the representations learned by neural networks.

While we find that non-robust neural networks do not strongly support the universality hypothesis, there is a convergence between both the representations used and the functions encoded by robust neural networks of different architectures. If true, even in the limited case of robust models, the universality hypothesis has substantial implications for the field of machine learning, and more broadly artificial intelligence and neuroscience. First, identifying and understanding the representations used by any individual neural network may allow us to understand the representations learned by *every* neural network that has been trained on the same dataset. This can have applications in mitigating

transferable adversarial examples [Moosavi-Dezfooli et al., 2017] as well as building representations that are more useful for transfer learning [Salman et al., 2020]. Second, if architecture matters less given a robustness constraint, robust representations may give us insight into patterns learned by biological brains [Conwell et al., 2021a,b, Zhuang et al., 2021, Yamins et al., 2014, Güçlü and van Gerven, 2015, Eickenberg et al., 2017].

## Acknowledgements

The authors would like to thank Jacob R. Gardner for his valuable feedback that inspired the stitching experiments. This research used resources provided by the Darwin testbed at Los Alamos National Laboratory (LANL) which is funded by the Computational Systems and Software Environments subprogram of LANL's Advanced Simulation and Computing program (NNSA/DOE). This work was supported by the Laboratory Directed Research and Development program of LANL under project number 20210043DR. LANL is operated by Triad National Security, LLC, for the National Nuclear Security Administration of the U.S. Department of Energy (Contract No. 89233218CNA000001).

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
