# OpenReview forum: "If You've Trained One You’ve Trained Them All: Inter-Architecture Similarity Increases With Robustness"
_auai.org/UAI/2022/Conference — UAI 2022 Oral_

### Official Review · Reviewer_heAc · 2022-04-04

**Q2(1) Originality/Novelty:** 4
**Q2(2) Significance/Impact:** 4
**Q2(3) Correctness/Technical Quality:** 4
**Q2(6) Clarity Of Writing:** 4
**Q6 Overall Score:** 8
**Q8 Confidence In Your Score:** 3

**Q1 Summary And Contributions:**

The paper proposed to reinvestigate prior works on analysis of neural network classifier similarity in light of adversarial robustness. Whereas prior works have provided substantial empirical evidence to indicate that there is a high correlation between models, the paper under review delves deeper into this aspect and experimentally finds it to be largely a consequence of activations being shared on natural images. (out of characters on open review... short summary ....)

**Q2 Assessment Of The Paper:**

More detailed information regarding each of these aspects is given below:

**Q2(4) Quality Of Experiments (Optional):**

4: Excellent: The experimental evaluation is comprehensive and the results are compelling.

**Q2(5) Reproducibility:**

2: Fair: Key resources (e.g., proofs, code, data) are unavailable but key details (e.g., proof sketches, experimental setup) are sufficiently well-described for an expert to confidently reproduce the main results.

**Q3 Main Strengths:**

The paper displays an interesting line of arguments and provides an insightful empirical analysis into model similarity in light of adversarial robustness. The paper is very well written, nicely structured, hypotheses and related works are clearly illustrated, and findings are easy to follow. This makes the paper a pleasure to read and although some might argue that the paper is simple in nature (it does not propose any new algorithms, leverages existing techniques, and is mainly of empirical form), this simplicity also summarizes its appeal. As also visible from the related works being published in top-tier venues, I concur that it is a good idea to publish these types of empirical works with clear motivation and analysis. They generally help advance our community’s intuition, while critically and constructively questioning the extent of previously obtained insights from prior literature.

**Q4 Main Weakness:**

As the only concern I believe that more effort should be invested into assuring reproducibility. In principle the paper’s insight should be replicable, as much of the baseline work is provided through the third-party pre-trained repository (table 1)  and other techniques are referenced respectively. However, there have been concerns over such practice not necessarily resulting in reproducible research and we should rule out that various other factors come into play in these analysis (specific random seeds, code versions, left-over bugs, etc.). A detailed account of precise set-ups in an appendix and the availability of code would be appreciated.

**Q5 Detailed Comments To The Authors:**

* This isn’t really a weakness, but the references are inconsistent and should be fixed. For instance, sometimes URLs are provided, sometimes not, sometimes conferences are indicated as proceedings, sometimes not, sometimes locations are added… etc.
* As a future suggestion to the authors and a complementary aspect to the adversarial robustness question, it would be interesting to see how the conducted analysis plays out if “dataset distillation” (Wang 2018) or “dataset condensation” (Zhao 2020) are investigated, in addition to the inverted dataset question.

**Q7 Justification For Your Score:**

I did not find any main technical weakness, other than perhaps that it is a largely empirical paper. However, as stated above, I believe the experimental analysis to be well motivated, rigorous and the insights to be discussed factually. As such, I have little concerns with respect to potential over-claims. One could naturally always conjecture that more experiments are necessary, but in terms of being acceptable for publication I believe the manuscript is sufficiently detailed already.

**Q9 Complying With Reviewing Instructions:**

1: Yes.

---

### Official Review · Reviewer_xRPm · 2022-04-10

**Q2(1) Originality/Novelty:** 2
**Q2(2) Significance/Impact:** 2
**Q2(3) Correctness/Technical Quality:** 2
**Q2(6) Clarity Of Writing:** 3
**Q6 Overall Score:** 6
**Q8 Confidence In Your Score:** 3

**Q1 Summary And Contributions:**

This papers studies the similarity between neural networks by comparing with the neuron activations. This paper proposes to investigate and calculate the similiarity using centered kernel alignment, representational-layer inversion and representation stitching. Based on the problem analysis of current similarity measures, this paper demonstrate the relations between robust neural networks and non-robust neural networks, showing there are common properties shared between robust neural networks.

**Q2 Assessment Of The Paper:**

More detailed information regarding each of these aspects is given below:

**Q2(4) Quality Of Experiments (Optional):**

2: Fair: The experimental evaluation is weak: important baselines are missing, or the results do not adequately support the main claims.

**Q2(5) Reproducibility:**

2: Fair: Key resources (e.g., proofs, code, data) are unavailable but key details (e.g., proof sketches, experimental setup) are sufficiently well-described for an expert to confidently reproduce the main results.

**Q3 Main Strengths:**

1. This paper gives detail analysis about the similarity between nerual networks, with regard to the learned neuron activations. This paper points out the overestimation problem of network similarity with empirical risk minimization, and shows that this problem can lead to errors on large scale datasets like image data.
2. This paper draws connection between robust and non-robust neural networks, with regard to the similartity between representations. The results show insights that the similarity between representations are highly related with the data itself, rather than the network structure or optimization mehtod.
3. The analysis based on similarity measure contributes well to the adversarial robustness performance.

**Q4 Main Weakness:**

1. This paper proposes to use gradient descent to find image pairs that contribute to the similarity measure with image features. However, it is not clearly represented that how these image features are derived, and how to link these features to the robustness of neural networks.
2. This paper shows insights on how the similarity measure are connected with the adversarial robustness, but it is not clear how this relation can be connected with the learning of adversarial functions.

**Q5 Detailed Comments To The Authors:**

One of the main contribution of this paper is to show insights that robust neural networks with different architectures like Densenet, Resnet have high similarity, indicating that the representations learned by robust networks are more generalizable. It might be better to support this point by some additional experiments, e.g., the representation stitching method proposed in section 3.

**Q7 Justification For Your Score:**

This paper studies an interesting and important problem, and provides some insights on the performance of similarity measures and the problem of overestimation. These insights can be related to the performance on adversarial robustness tasks, which is important to many maching learning applications. Based on these contributions, I conclude with the above overall score.

**Q9 Complying With Reviewing Instructions:**

1: Yes.

---

### Official Review · Reviewer_F4As · 2022-04-13

**Q2(1) Originality/Novelty:** 3
**Q2(2) Significance/Impact:** 3
**Q2(3) Correctness/Technical Quality:** 3
**Q2(6) Clarity Of Writing:** 3
**Q6 Overall Score:** 7
**Q8 Confidence In Your Score:** 4

**Q1 Summary And Contributions:**

The paper shows that adversarially robust neural networks are more similar to each other than networks without adversarial training, even if architectures are different. This is done in two ways: (1) with an inverted dataset, to measure how similar two models' activations are; and (2) by stitching together networks.

**Q2 Assessment Of The Paper:**

More detailed information regarding each of these aspects is given below:

**Q2(4) Quality Of Experiments (Optional):**

4: Excellent: The experimental evaluation is comprehensive and the results are compelling.

**Q2(5) Reproducibility:**

2: Fair: Key resources (e.g., proofs, code, data) are unavailable but key details (e.g., proof sketches, experimental setup) are sufficiently well-described for an expert to confidently reproduce the main results.

**Q3 Main Strengths:**

* The experiments support the conclusion from multiple angles
* The paper is easy to understand.

**Q4 Main Weakness:**

* The "inverted dataset" is constructed adversarially, and may not really represent the true dataset. Models that are adversarially trained might have different behavior for this inversion process, since that is exactly what they are trained against. So the conclusions about adversarial training might be an artifact of this dataset construction
* The experiments are limited to a single dataset, so the conclusion might not generalize
* The neural networks are all convolutional, with similar architectures
* Source code is not provided

**Q5 Detailed Comments To The Authors:**

* Table 1 is useless, it just shows which datasets are used in the other figures.
* Figure 7: this measures agreement between the stitched model and the head model, but what about agreement with the body model? Perhaps if head and body disagree, the stitched model sometimes agrees with the body model instead.
* I wonder if other architectures, such as vision transformers, behave similarly with robust training.
* Clarify that ||_F in equation (1) denotes the Frobenius norm
* "Suppose that two features are perfectly correlated in a dataset .. for some |c| > 0" \
  Why not say "for some c ≠ 0"?

* spelling errors:
  * "generalizeable" -> "generalizable"
  * "present the results in in Figure 4.": double "in"
  * "agreee" -> "agree"


**Q7 Justification For Your Score:**

An interesting finding that is well analyzed.

**Q9 Complying With Reviewing Instructions:**

1: Yes.

---

### Decision · Program_Chairs · 2022-05-15

**Decision:**

Accept (Oral)

**Comment:**

Meta Review: The paper investigates similarity between representations learned by different neural networks on the same task. The paper presents two main findings:
* Previously used methods to quantify representation similarity based on feature correlations (for the same inputs) have a flaw that can be explained theoretically, and leads these methods to significantly overestimate network similarity
* The paper proposes an alternative approach, that confirms the overestimation of similarity of standard networks, and finds that increasing the robustness of networks to adversarial inputs increases similarity of representations (under the new measure).

The paper addresses a timely and important problem: comparing representations learned by networks is a notoriously tricky problem, that plays an important role for downstream explanation and analysis across a range of works. Pointing out a flaw (and overestimation of similarity) is an significant and original contribution that deserves a publication on its own. On top of that, the paper shows that adversarial robustness correlates with feature similarity; providing inspiration or future theoretical explanations that further the community's understanding of (robust) representations. All reviewers argue in favor of accepting the paper, and I want to second that verdict.

Pro:
* Timely topic, and important main findings, with potential impact for wider parts of the community
* Clear main arguments that are well presented, clearly articulated, and easy to follow
* Well executed main experiments that support conclusions very well

Con:
* Generality of findings somewhat limited (conv nets and single dataset)
* Some issues with reproducibility raised by reviewers, but authors promise to address them in the camera ready version